# Tissue and cellular tropism of elephant endotheliotropic herpesvirus (EEHV)1A in hemorrhagic disease

Jennifer A. Landolfi [1]*, Lauren Howard [2], Paul Ling [3]

1 Zoological Pathology Program, University of Illinois, Brookfield, Illinois, United States of America, 2 Peel Therapeutics, Salt Lake City, Utah, United States of America, 3 Department of Virology and Microbiology, Baylor College of Medicine, Houston, Texas, United States of America

☯ These authors contributed equally to this work.
* landolfi@illinois.edu

## Abstract

Elephant endotheliotropic herpesviruses (EEHVs) cause EEHV hemorrhagic disease (EEHV-HD), an acute, multisystemic, often fatal hemorrhagic syndrome with profound implications for elephant population growth and sustainability. A greater understanding of the pathogenesis of EEHV-HD is essential to elucidate susceptibility and develop tools for disease management and prevention. This study utilized RNAscope® *in situ* hybridization (ISH) to detect EEHV1A DNA polymerase and terminase genes in archival tissues (heart, lung, tongue, spleen, liver, kidney, lymph node, stomach, small intestine, large intestine, salivary gland, and brain or spinal cord) from Asian elephants (*Elephas maximus*; n = 12) that died of EEHV-HD to determine and describe tissue and cellular tropism of the virus. Tissue and cellular specific ISH signal were recorded and semi-quantitatively graded using light microscopy. Positive hybridization signal for EEHV1A terminase and DNA polymerase was detected in tissues from all twelve study cases. In all tissues, positive signal was limited to endothelial cell nuclei. No signal was detected in epithelial cells, leukocytes or mesenchymal cells other than endothelial cells. Signal detection frequency was as follows: heart (12/12), liver (11/12), tongue (10/12), lymph node (10/12), spleen (9/11), stomach (9/12), small intestine (9/10), large intestine (9/10), lung (7/10), salivary gland (1/8), kidney (1/12), brain/spinal cord (0/10). Tissue signal amount varied among cases but generally was most abundant in heart and liver. Results confirmed that in Asian elephant cases of EEHV1A-HD, the viral cellular target and site of replication is exclusively capillary endothelial cells. Differences in viral tissue tropism with EEHV1A-HD are likely a consequence of endothelial cell heterogeneity across tissues. Understanding tropism in cases of active EEHV-HD can serve as a foundation for investigation of EEHV tropism in other stages of the infection (e.g. initial infection, dissemination, latency, shedding) and contribute to defining pathogenesis.

**Data availability statement:** All relevant data are within the paper and its Supporting Information files.

**Funding:** The research was funded by a grant from the American Association of Zoo Veterinarians Wild Animal Health Fund (AAZV Proposal: 2022 #4). • The funders had no role in study design, data collection and analysis, decision to publish, or preparation of the manuscript.

**Competing interests:** The authors have declared that no competing interests exist.

## Introduction

Elephant endotheliotropic herpesviruses (EEHVs) cause EEHV hemorrhagic disease (EEHV-HD), a condition responsible for mortality in young Asian (*Elephant maximus*) and African (*Loxodonta africana*) elephants. In Asian elephants born since 1980 in North America and Europe, EEHV-HD is the single greatest cause of death. The disease is an acute, multisystemic, often fatal hemorrhagic syndrome [1–11]. More than 50 confirmed fatal cases of EEHV-HD have occurred in North American and European zoos. Additionally, more than 140 cases have been confirmed via PCR in free-ranging, managed, and working elephant calves in range countries [5,9,11]. Infectious disease, in general, can be a significant threat to both *ex* and *in situ* conservation efforts and species survival. Considering most EEHV-HD cases have occurred in calves, infection and disease have profound implications for elephant population growth and sustainability. A greater understanding of the pathogenesis of EEHV-HD is essential to elucidate susceptibility and develop tools for disease management and prevention.

Several species of EEHVs (genus *Proboscivirus)* exist, and multiple strains of each EEHV species can cause disease. *Proboscivirus* infection is ubiquitous in adult, clinically normal elephants. EEHV1A, EEHV1B, EEHV4 and EEHV5 are endemic in Asian elephant populations, while EEHV2, EEHV3, EEHV6, and EEHV7 are endemic in African elephants [12]. Most cases of lethal EEHV-HD in Asian elephants are due to EEHV1A [1,7,8]. Asian elephant calves, typically between 2–8 years of age, are most susceptible to EEHV-HD. Maternal antibody appears to provide protection against disease given the absence of EEHV-HD in animals less than one year of age. Research has also demonstrated that EEHV1 antibodies wane in Asian elephant calves during the first two years of life [13]. Importantly, serologic evidence indicates that EEHV-HD results from primary infection rather than reactivation of latent virus [13,14]. Though more needs to be learned about the mechanisms governing susceptibility to EEHV-HD, current knowledge indicates that significant EEHV-related disease develops when primary infection of a naïve calf is not controlled by normal innate cellular and humoral immune responses [5,13].

Lesions of EEHV-HD are consistent with endothelial cell damage and vascular compromise with resultant hemorrhage, edema, and coagulopathy. Targeting of endothelial cells is indicated by the presence of herpesviral intranuclear inclusion bodies in these cells. Direct virus-mediated injury is considered the most likely mechanism of endothelial cell damage though host immune responses likely also contribute. Death is the result of hypovolemic shock due to cardiac and circulatory collapse [5,15].

Though lesions provide strong evidence for endothelial cell tropism during EEHV-HD, it is unlikely that endothelial cells are the site of primary infection, vehicle for systemic dissemination, or source of viral shedding. Tropism is one of the features that distinguishes herpesvirus subfamilies and defines manifestations of disease. For EEHVs, the cell types that participate in initial infection, systemic dissemination, development of hemorrhagic disease lesions and viral latency have not been well

defined. Determination of viral tissue and cellular tropism has the potential to improve understanding of these disease dynamics and correlate infection with host immunity.

To establish tissue and cellular tropism, techniques, such as *in situ* hybridization (ISH) and immunohistochemistry (IHC), that can facilitate visualization of virus in tissue are required. To date, few studies have utilized IHC or ISH to study viral tropism and distribution in archival cases [16–18]. Additional work is necessary to build upon findings. The current study utilized RNAscope® ISH technology to detect EEHV1A in archival tissues from Asian elephants that died of EEHV-HD to determine and describe viral tissue and cellular tropism reflective of productive infection/viral replication during active disease. Because tropism provides the basis for infectivity, dissemination, shedding and latency, its definition is the first step in understanding pathogenesis and impacting susceptibility.

## Materials and methods

### Samples

Formalin-fixed, paraffin-embedded (FFPE) archival tissues from PCR-confirmed Asian elephant cases of EEHV-HD due to EEHV1A (N = 12) and elephants (one Asian and one African) that died due to other causes (N = 2; negative controls) were obtained. EEHV PCR testing was done at the time of initial clinical diagnosis or retrospectively by the National Elephant Herpesvirus Laboratory, Smithsonian's National Zoo. From each case (per availability), heart, lung, tongue, spleen, liver, kidney, lymph node, stomach, small intestine, large intestine, salivary gland, and brain or spinal cord were analyzed (Table 1).

### *In situ* hybridization

Five um thick, serial sections of each tissue from each case were cut and mounted on RNase-treated, charged glass slides. In conjunction with Advanced Cell Diagnostics (ACD), molecular probes designed to hybridize to the DNA polymerase and terminase genes of EEHV1A were synthesized based on published sequence information (GenBank Accession KC618527; PN808191 and PN808201, ACD, Newark, CA). A probe targeting Asian elephant beta actin (GenBank Accession FJ423082.1; PN809231, ACD) was synthesized to serve as a protocol positive control [19]. All tissue sections were hybridized with all three probes and a negative control probe (Negative control probe dapB, PN310043, ACD). Chromogenic ISH was carried out according to manufacturer's instructions (RNAscope 2.5 HD Detection Reagent-Red Kit, PN 322360, ACD; [20]). Briefly, FFPE slides were baked at 60°C for 1 hour, deparaffinized in xylene for 10 minutes (x

**Table 1. Elephant tissues evaluated for EEHV1A genes (DNA polymerase and terminase) via RNA Scope® *in situ* hybridization.**

| Case number | Heart | Liver | Tongue | Lymph node | Salivary gland | Spleen | Kidney | Stomach | Small intestine | Large intestine | Lung | Brain/Spinal cord |
|---|---|---|---|---|---|---|---|---|---|---|---|---|
| 1 | X | X | X | X | – | X | X | X | X | X | – | X |
| 2 | X | X | X | X | X | X | X | X | X | X | X | X |
| 3 | X | X | X | X | X | X | X | X | X | X | X | – |
| 4 | X | X | X | X | X | X | X | X | X | X | X | – |
| 5 | X | X | X | X | X | X | X | X | X | X | – | X |
| 6 | X | X | – | X | – | X | X | X | X | X | X | X |
| 7 | X | X | – | X | X | – | X | X | X | X | X | X |
| 8 | X | X | X | X | X | X | X | X | X | X | X | X |
| 9 | X | X | X | X | – | X | X | X | – | – | X | X |
| 10 | X | X | X | X | X | X | X | X | X | – | X | X |
| 11 | X | X | X | X | – | X | X | X | – | X | X | X |
| 12 | X | X | X | X | X | X | X | X | X | X | X | X |

2), rinsed in 100% ethanol twice, air-dried, and then rehydrated with dH2O for 2 minutes. "Pretreatment solution 1" was applied to the slides for 10 minutes at room temperature. The slides were then boiled in "Pretreatment solution 2" at 100°C for 30 minutes, followed by protease digestion in "Pretreatment solution 3" for 30 minutes at 40°C to allow target accessibility. Specific or control probes were applied, and the slides were incubated at 40°C for 2 hours. Slides were washed twice with 1X wash buffer for 2 minutes at room temperature. A series of 6 signal amplification steps were performed per manufacturer's instructions. Incubation with RED A and B solutions was performed at room temperature for 10 minutes, and the reaction was quenched with dH2O. Gill's hematoxylin was applied for 2 minutes, and slides were then rinsed in water, decolorized in 0.02% ammonium hydroxide, and rinsed in water again. Slides were passed through 100% ethanol, 70% ethanol, and twice in xylene before cover slipping with xylene-based mounting medium.

Slides were examined with a light microscope by one of the authors, an American College of Veterinary Pathologists (ACVP) board certified pathologist (JAL). Sample viability was first assessed via evaluation of control beta actin hybridized slides. Negative control hybridized sections were examined to ensure signal specificity. EEHV DNA polymerase and terminase gene hybridized samples were then evaluated for signal indicating the presence or absence of viral nucleic acid. Tissue and cellular specific signal were recorded and analyzed for each section in order to establish EEHV1A tropism. The average amount of signal detected in ten 200x magnification fields was determined. Signal was semi-quantitatively graded as follows: Grade 1–1–9 positive nuclei/200x magnification field; Grade 2–10–20 positive nuclei/200x field; Grade 3–21–30 positive nuclei/200x field; Grade 4 – greater than 30 positive nuclei/200x field (S1 Fig).

## Results

Heart, liver, lymph node, kidney, and stomach were available for analysis in all 12 EEHV1A-HD cases. Total cases for which other tissues were available were as follows: tongue (n = 10), salivary gland (n = 8), spleen (n = 11), small intestine (n = 10), large intestine (n = 10), lung (n = 10), brain or spinal cord (n = 10; Table 1). All tissues were assayed for the two negative control cases. In all cases, tissues hybridized with the negative control probe lacked hybridization signal and were free of nonspecific background staining (S2 Fig). Tissues from all cases had positive hybridization signal for Asian elephant beta actin. Distribution and intensity of beta actin signal varied among cases and tissues. Cytoplasmic signal was common in vascular and alimentary tract smooth myocytes, neurons and glial cells, leukocytes and various epithelial cells. Nuclear signal was also sometimes prominent in cardiac, skeletal and smooth muscle and endothelial cells (S3 Fig).

Positive hybridization signal for EEHV1A terminase and DNA polymerase was detected in tissues from all twelve study cases of fatal EEHV-HD PCR confirmed to be due to EEHV1A; no signal was detected in tissues from the two negative control cases (Table 2). Tissue and cellular distribution of hybridization signal was similar in all EEHV1A-HD cases, though variation in extent of tissue involvement and amount of signal within tissues was noted (Fig 1). Signal distribution and intensity for terminase and DNA polymerase probes were equivalent and thus described as one. In all tissues, definitive positive hybridization signal was present in the nuclei of spindaloid, mesenchymal cells lining vascular spaces (morphologically consistent with endothelial cells). Hybridization signal ranged from a saturated deep red that filled the nucleus and often seeped beyond the nuclear membrane margins to intranuclear punctate, red to pink foci (Fig 2).

In all cases for which the tissues were available for assay, positive hybridization signals for both terminase and DNA polymerase were detected in heart (12/12) and tongue (10/12). In the heart, capillary endothelial cells throughout the myocardium had hybridization signal (Fig 3). No signal was detected in cardiomyocytes. A range of endothelial cell signal amount was evident in the study cases. Grade 4 signal was observed in two cases; grade 3 in four; grade 2 in four; and grade 1 in two. In the tongue, hybridization signal was evident in capillary endothelial cells throughout the glossal skeletal muscle and ranged from grade 1 (6/10) to grade 2 (4/10). No signal was detected in glossal mucosal epithelium or skeletal myocytes. Most cases (11/12) had positive hybridization for EEHV1A genes in sections of liver. Signal was detected in nuclei of sinusoidal endothelial cells (Fig 4). No signal was detected in hepatocytes, biliary epithelium or Kupffer cells. The amount of hepatic sinusoidal endothelial cell signal varied widely among cases (grade 1 = 3, grade 2 = 5, grade 3 = 1,

**Table 2. RNA Scope® *in situ* hybridization signal grade for EEHV1A genes (DNA polymerase and terminase) in elephant tissues.**

| Case number | Heart | Liver | Tongue | Lymph node | Salivary gland | Spleen | Kidney | Stomach | Small intestine | Large intestine | Lung | Brain |
|---|---|---|---|---|---|---|---|---|---|---|---|---|
| 1 | 3 | 3 | 2 | 1 | NA | 1 | 0 | 1 | 1 | 1 | NA | 0 |
| 2 | 4 | 2 | 2 | 1 | 0 | 1 | 0 | 1 | 2 | 2 | 1 | 0 |
| 3 | 3 | 2 | 1 | 1 | 0 | 2 | 1 | 1 | 1 | 0 | 1 | NA |
| 4 | 1 | 1 | 1 | 0 | 0 | 0 | 0 | 1 | 1 | 1 | 0 | NA |
| 5 | 2 | 2 | 1 | 1 | 0 | 1 | 0 | 1 | 1 | 1 | NA | 0 |
| 6 | 4 | 4 | NA | 1 | NA | 2 | 0 | 1 | 1 | 1 | 1 | 0 |
| 7 | 2 | 1 | NA | 1 | 0 | NA | 0 | 1 | 1 | 1 | 0 | 0 |
| 8 | 3 | 1 | 1 | 1 | 1 | 2 | 0 | 1 | 1 | 1 | 1 | 0 |
| 9 | 2 | 2 | 2 | 2 | NA | 2 | 0 | 0 | NA | NA | 1 | 0 |
| 10 | 1 | 0 | 1 | 0 | 0 | 0 | 0 | 0 | 0 | NA | 0 | 0 |
| 11 | 4 | 2 | 1 | 1 | NA | 2 | 1 | 0 | NA | 1 | 1 | 0 |
| 12 | 3 | 4 | 2 | 1 | 0 | 1 | 0 | 1 | 1 | 1 | 1 | 0 |

Signal was semi-quantitatively graded as follows: Grade 0 – no positive nuclei; Grade 1–1–9 positive nuclei/200x magnification field; Grade 2–10–20 positive nuclei/200x field; Grade 3–21–30 positive nuclei/200x field; Grade 4 – greater than 30 positive nuclei/200x field. NA, tissue not available for assay.

grade 4 = 2). In all other examined tissues with positive hybridization signal, signal was detected in the nucleus of cells morphologically compatible with capillary endothelial cells, and amount ranged from grade 1 to 2. In 10/12 cases, signal was detected in lymph node (grade 1 = 9/10, grade 2 = 1/10; Fig 5), and in 9/11 cases, it was detected in spleen (grade 1 = 4/9, grade 2 = 5/9). No definitive signal was detected in mononuclear round cells morphologically consistent with leukocytes present in either tissue. Signal was detected in sections of alimentary tract (stomach 9/12, small intestine 9/10, large intestine 9/10). In these tissues, signal was most frequent in the capillary endothelial cells of the muscularis and very rare in the submucosa and mucosal lamina propria. Signal was grade 1–2 in all cases. No signal was detected in epithelial cells or smooth myocytes of the alimentary tract. Signal was detected in capillary endothelial cells of the lung in 7/10 cases and was grade 1 in all instances. No signal was detected in pulmonary epithelial cells. In the eight cases for which salivary gland was available for assay, only one had capillary endothelial cell signal (grade 1). No signal was detected in salivary gland epithelium. In sections of kidney, only 1/12 cases had capillary endothelial cell signal (grade 1). No signal was detected in renal tubular or glomerular epithelial cells or glomerular endothelial cells. No signal was detected in brain or spinal cord from cases for which tissue was available for assay (n = 10).

## Discussion

Results of the study confirmed that in Asian elephant cases of EEHV1A-HD, the viral cellular target was capillary endothelial cells. Detection of EEHV1A DNA polymerase and terminase genes, genes required for virus generation, specifically provided evidence of viral replication in the infected endothelial cells as a hallmark of EEHV-HD. Virus tissue distribution was not uniform or even across examined samples. Capillary endothelial cells of heart and liver had elevated levels of ISH signal as compared with other evaluated tissues. Importantly, no signal was detected in salivary gland or gastrointestinal epithelium, central nervous system tissues or leukocytes. Findings indicated that the stage of EEHV1A infection defined as hemorrhagic disease was characterized by viral infection of/replication in capillary endothelial cells exclusively. Active, replicating infection of other cell types was not detected.

Several recent studies have advanced the understanding of EEHV-HD pathogenesis. Investigators have detected viral antigens in epithelia of the alimentary tract and salivary glands indicating the possibility of initial epithelial infection at these sites [16,18]. Identification of viral antigen-containing blood monocytes has suggested a means of dissemination

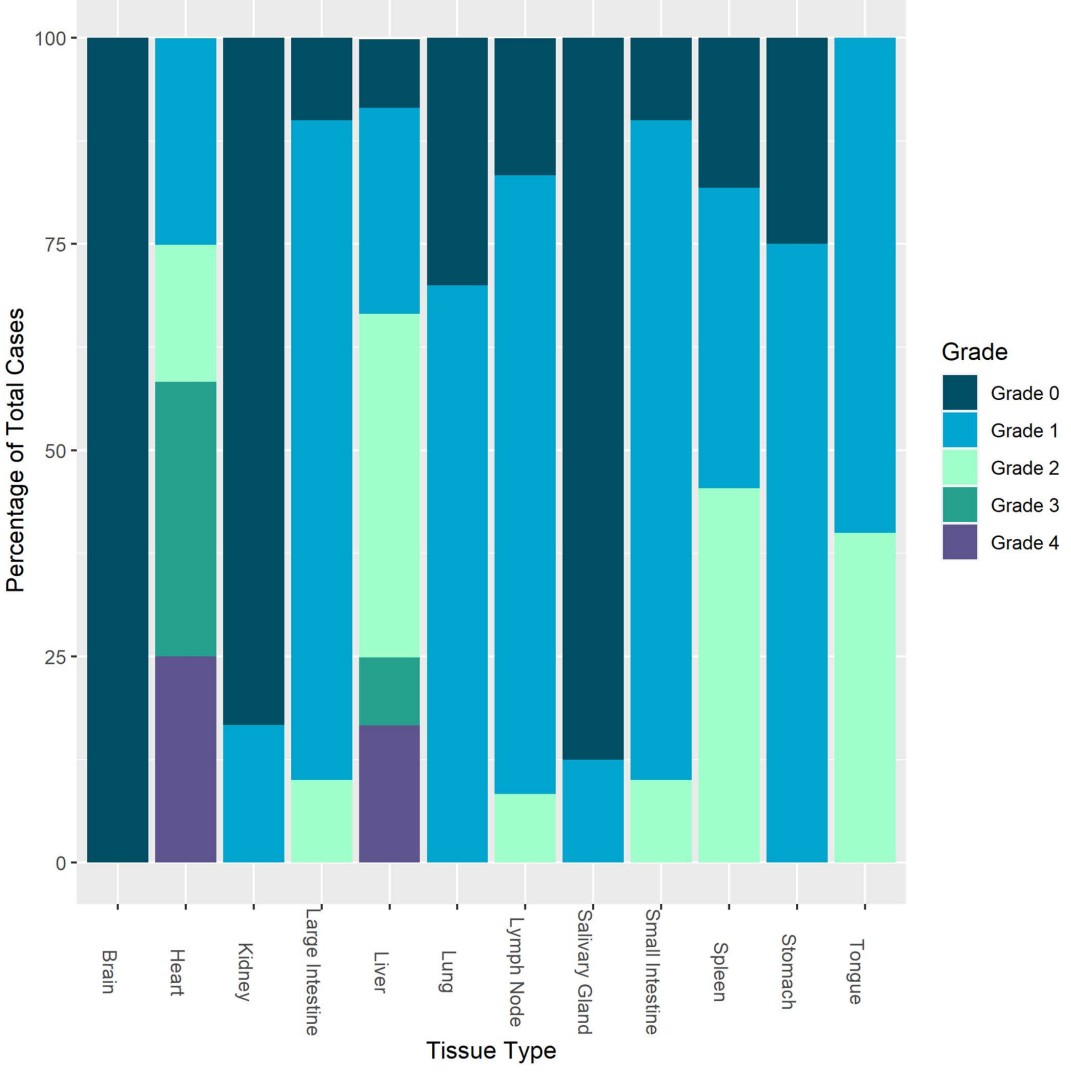

**Fig 1. RNA Scope® *in situ* hybridization signal grade for EEHV1A genes (DNA polymerase and terminase) in elephant tissues.** For each examined tissue (x axis), RNA Scope® *in situ* hybridization signal is stratified according to grade and reported as the percentage of total cases (y axis) for which the particular tissue was available for assay.

[16–18,21]. Finally, research has demonstrated that transmission of infection to vascular endothelial cells is facilitated by monocyte-leukocyte adhesion molecules (i.e. PCAM-1; [16]). Results of these previous investigations appear to disagree with the current study where virus was only detected in capillary endothelial cells. It is important to note, however, that previous studies described positive signal/labeling against the DNA polymerase gene/protein in the cytoplasm and not the nucleus of epithelial cells and monocytes [16,17]. EEHV is a DNA virus that replicates in the nucleus of infected cells where the DNA polymerase protein also localizes; thus, cytoplasmic location of virus/viral antigen would be most suggestive of non-productive infection. Non-productive infection could be a feature of initial epithelial invasion and dissemination of infection with EEHV; however, findings of the current study show that clinical hemorrhagic disease is characterized exclusively by viral replication within and consequent damage to capillary endothelial cells. Evidence of epithelial invasion/involvement and/or dissemination via leukocyte trafficking was not detected in the cases of active EEHV-HD. In the future,

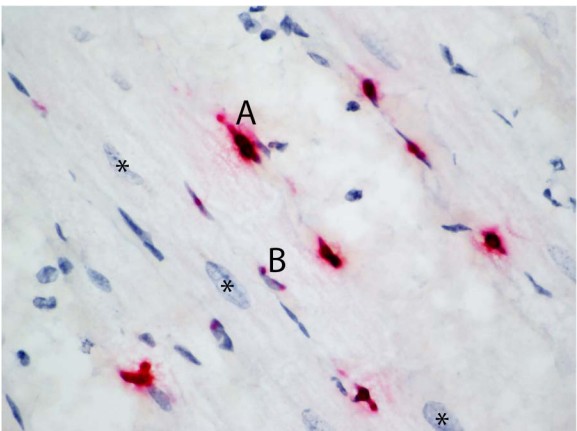

**Fig 2. Variation in endothelial cell RNA Scope® *in situ* hybridization signal intensity.** Myocardium from Case 2 hybridized with the EEHV1A terminase probe. Saturated deep red signal that fills the nucleus and seeps beyond the nuclear membrane margins of the endothelial cell is exemplified by A; intranuclear punctate, red to pink foci of signal are exemplified by B. Asterisks mark cardiomyocyte nuclei.

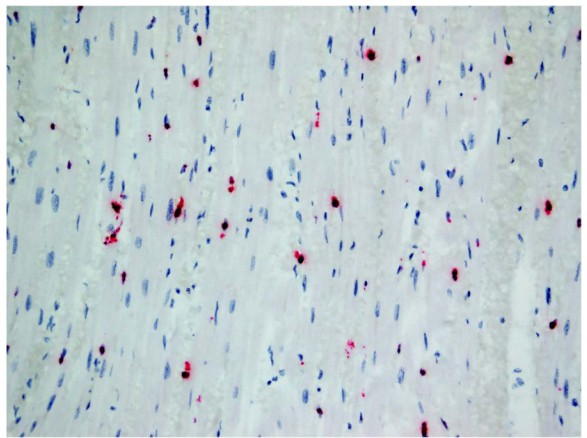

**Fig 3. EEHV1A terminase gene RNA Scope® *in situ* hybridization on heart from Case 1.**

studies providing specific detection of EEHV DNA genomes within cells in the absence of viral lytic gene RNA or protein expression might clarify the proportion of nonproductively infected cells to cells with ongoing active replication and further elucidate viral tropism.

Experimental methodology was another factor that could have influenced discrepancies in results between the current and previous studies. Prior studies utilized immunohistochemistry or conventional ISH to detect virus in archival tissue samples. The current project utilized a newer, novel ISH technique (RNAscope®) that offers enhanced specificity and sensitivity of detection as compared with conventional ISH and IHC protocols. With RNAscope®, amplification of target signal occurs via a cascade of hybridization events. Molecular probes (as opposed to antibody-based detection) are utilized and designed to ensure selective amplification of target-specific signals and to avoid background. Another advantage is amplification of a relatively short target region that facilitates hybridization of partially degraded nucleic acid targets; degradation may be a frequent occurrence in archival tissue samples and cause of reaction failure in conventional assays.

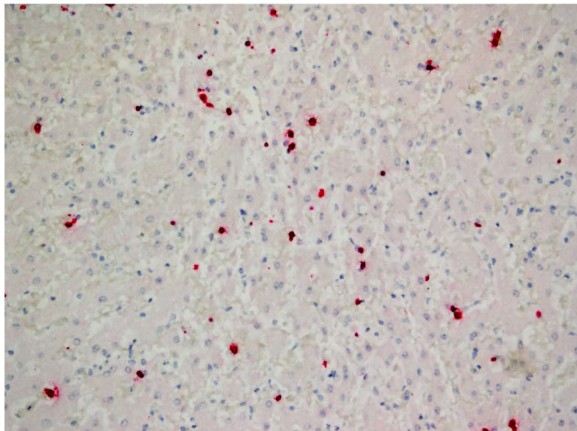

**Fig 4. EEHV1A terminase gene RNA Scope®** *in situ* **hybridization on liver from Case 6.**

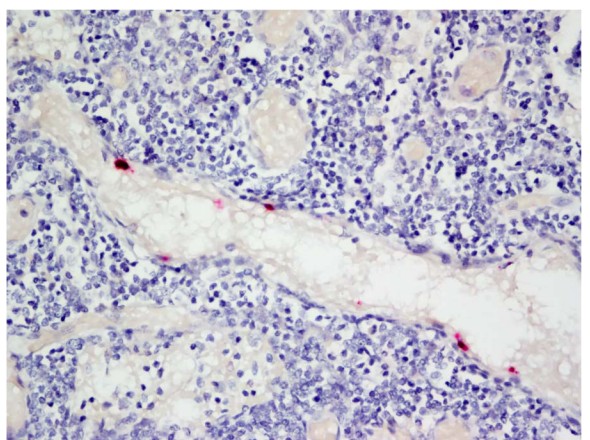

**Fig 5. EEHV1A terminase gene RNA Scope®** *in situ* **hybridization on lymph node from Case 8.** Endothelial cells lining the vascular profile have positive (red) hybridization signal in the nucleus. Surrounding mononuclear round cells (leukocytes) have no signal.

Altogether, these features inherent to the RNAscope® technology could have provided improved performance and results as compared with conventional assays.

Another interesting finding highlighted by the results of the study was the differences in tissue tropism for EEHV1A. In all tissues, capillary endothelial cells were the only cells with positive EEHV1A ISH signal, but distribution of signal across tissues and amount of signal within tissues varied. Signal was detected in the heart in all cases and was generally most abundant at this location followed by liver. These were the only two tissues for which a signal of grade 3 or 4 was detected. In all other evaluated tissues with positive ISH signal, only signals of grades 1 or 2 were noted. Differences in viral tissue tropism with EEHV1A-HD were likely a consequence of endothelial cell heterogeneity across tissues. Endothelial cells are the most ubiquitous cells of the body lining all inner surfaces of the cardiovascular system and perform a variety of vital protective, receptor, transport, secretion, absorption, and synthesis functions. Importantly, endothelial cells are not uniform and display unique phenotypes defined by the tissue and vascular locations they inhabit. The cells have tissue-specific phenotypes that are modified along the length of the vascular bed to accommodate essential functions.

Additionally, endothelial cells can adapt to environmental cues through the mobilization of genetic, molecular, and structural alterations adding an element of plasticity to their inherent phenotypic heterogeneity [22–25]. Unique structural and/or physiologic features afforded to endothelial cells of specific locations may lead to preferential EEHV1A infection of these cells resulting in the specific tissue tropism illustrated in the study.

Given that positive ISH signal was only detected in capillary endothelial cells in all evaluated samples, the current study was unable to elucidate mechanisms of pathogenesis such as initial infection, dissemination, shedding and latency. Capillary endothelial cell infection and replication appeared to be vital features of the hemorrhagic disease stage of EEHV1A infection. At this stage of the disease and in these fatal infections, no ISH signal distribution compatible with active viral shedding or sites of latency were observed. This shortfall of the study may be attributable to disease stage; all study cases were fatal EEHV1A-HD cases. Results supported that active EEHV-HD, manifested by DNA polymerase and terminase expression, was not concurrent with periods of viral shedding or latency. Because the DNA polymerase and terminase genes targeted for virus identification in the study were biased for detection of productive infection or replication, these gene targets may not have been ideal for defining tropism in cells involved with nonproductive stages of the infection. Future studies surveying elephant tissues from survivors of previous EEHV-HD and/or healthy, infected adults and detecting other non-replicative gene targets (e.g. viral glycoprotein B) may provide further information. No viral positive ISH signal was detected in tissues from either of the negative control cases assayed in this study to suggest they harbored replicating virus in the absence of disease.

Results of the current study can provide direction for future investigations into infection of Asian elephants with other EEHV species and into EEHV infections of African elephants. Additionally, details defined about tropism in cases of active EEHV-HD can serve as a foundation for investigation of EEHV tropism in other stages of the infection (e.g. initial infection, dissemination, latency, shedding). This work will be applicable to improvements in treatment and prevention. Identifying factors that influence susceptibility and devising methods to combat them is one route towards ensuring greater calf survival.

## Supporting information

**S1 Fig. Representative images of grades 1–4 RNA Scope® *in situ* hybridization signal.** A. Myocardium from Case 4 hybridized with the EEHV1A terminase probe. Less than 10 endothelial cell nuclei in the 200x magnification field have positive *in situ* hybridization (ISH) signal represented by red staining. Findings are consistent with grade 1 signal. B. Myocardium from Case 5 hybridized with the EEHV1A terminase probe. Greater than 10 but less than 21 endothelial cell nuclei in the 200x magnification field have positive ISH signal. Findings are consistent with grade 2 signal. C. Myocardium from Case 1 hybridized with the EEHV1A terminase probe. Greater than 20 but less than 30 endothelial cell nuclei in the 200x magnification field have positive ISH signal. Findings are consistent with grade 3 signal. D. Myocardium from Case 2 hybridized with the EEHV1A terminase probe. Greater than 30 endothelial cell nuclei in the 200x magnification field have positive ISH signal. Findings are consistent with grade 4 signal.
(TIF)

**S2 Fig. Negative control probe RNA Scope® *in situ* hybridization on heart from Case 2.** No positive (red) *in situ* hybridization signal is visible in cardiomyocytes, vascular smooth myocytes, endothelial cells or any other cells represented in the image.
(TIF)

**S3 Fig. Beta actin gene RNA Scope® *in situ* hybridization on heart from Case 2.** Positive (red) *in situ* hybridization signal is visible in multiple locations including nuclei of cardiomyocytes and cytoplasm of cardiomyocytes, endothelial cells, vascular smooth myocytes and some circulating leukocytes.
(TIF)

## Acknowledgments

Gretchen Anchor provided vital technical support for RNA Scope® assays. The authors are also grateful to zoos that provided case materials for the study and work tirelessly to provide optimal health and welfare to elephants in managed care, especially while battling EEHV-HD.

## Author contributions

**Conceptualization:** Jennifer Landolfi.

**Data curation:** Jennifer Landolfi, Lauren Howard, Paul Ling.

**Formal analysis:** Jennifer Landolfi.

**Funding acquisition:** Jennifer Landolfi, Lauren Howard, Paul Ling.

**Investigation:** Jennifer Landolfi, Paul Ling.

**Methodology:** Jennifer Landolfi, Paul Ling.

**Project administration:** Jennifer Landolfi, Lauren Howard.

**Resources:** Jennifer Landolfi, Lauren Howard, Paul Ling.

**Supervision:** Jennifer Landolfi.

**Validation:** Jennifer Landolfi.

**Writing – original draft:** Jennifer Landolfi.

**Writing – review & editing:** Lauren Howard, Paul Ling.

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
