## [Decision Letter · Decision Letter 0]

4 Jun 2025

Dear Dr. Landolfi,

Thank you for submitting your manuscript to PLOS ONE. After careful consideration, we feel that it has merit but does not fully meet PLOS ONE’s publication criteria as it currently stands. Therefore, we invite you to submit a revised version of the manuscript that addresses the points raised during the review process.

We look forward to receiving your revised manuscript.

Kind regards,

Graciela Andrei

Academic Editor

PLOS ONE

Journal Requirements:

2. Thank you for stating the following financial disclosure: [The research was funded by a grant from the American Association of Zoo Veterinarians Wild Animal Health Fund (AAZV Proposal: 2022 #4).]. 

4. Thank you for stating the following in the Acknowledgments Section of your manuscript: [The research was funded by a grant from the American Association of Zoo Veterinarians Wild Animal Health Fund (AAZV Proposal: 2022 #4). Gretchen Anchor provided vital technical support for RNA Scope® assays. The authors are also grateful to zoos that provided case materials for the study and work tirelessly to provide optimal health and welfare to elephants in managed care, especially while battling EEHV-HD.]

Please remove any funding-related text from the manuscript and let us know how you would like to update your Funding Statement. Currently, your Funding Statement reads as follows: [The research was funded by a grant from the American Association of Zoo Veterinarians Wild Animal Health Fund (AAZV Proposal: 2022 #4).]. 

Reviewers' comments:

Reviewer's Responses to Questions

**Comments to the Author**

1. Is the manuscript technically sound, and do the data support the conclusions?

Reviewer #1: Partly

Reviewer #2: Yes

2. Has the statistical analysis been performed appropriately and rigorously?

Reviewer #1: N/A

Reviewer #2: N/A

3. Have the authors made all data underlying the findings in their manuscript fully available?

Reviewer #1: Yes

Reviewer #2: Yes

4. Is the manuscript presented in an intelligible fashion and written in standard English?

Reviewer #1: Yes

Reviewer #2: Yes

Reviewer #1: This is a very interesting work. However, the authors need to make clear the following:

The title says "viral tissue" not sure what they mean with this term. My suggestion is to modify the title.

Although the pathology and tissue observation was made by a certified pathologist, the interpretation of data on the manuscript is difficult to follow. It might be better describe by tissue?

As the authors mention, signal grade was semi-quantitative, ¿were those based only on the pathologist criteria? because there´s no reference associated to this.

Is there any data about the sensibility/specificity of the RNAscope 2.5 HD Detection Reagent-Red Kit.

The authors need to include images of the negative controls used with the assay.

The authors need to include images of positive tissues to beta-actin. It is possible to count number of positive cells to beta actin and make some statistics with this number and those positive to the virus genes?

The number of positive cells in each animal were similar? Let's say, in large intestine 8 samples out of 10 were considered positive grade 3, if they count 21-30 positive cells. But there's a difference of 9 cells between.

Is it possible to use these numbers to include some statistics?

Data on table 1 and 2, it is possible to present results in a more friendly/visual format? Table 3 repeats what is shown on table 2.

Authors mention that archival tissues were from PCR-confirmed Asian elephant cases of EEHV-HD due to EEHV1A. All tested tissues were positive to PCR?

What is the rationale to select the DNA polymerase and terminase genes to design the hybridization probes? Include also the sequence of each probe on the document.

The discussion takes the main aspects on the paper, including the novel kit used, the genes selected to target in the tissues. The main type of cells (endothelial) where positivity was detected.

An important aspect that should be discussed is the tissue treatment before in situ hybridization.

Reviewer #2: 1. The study investigates the tissue and cellular tropism of Elephant Endotheliotropic Herpesvirus 1A (EEHV1A) in fatal cases of haemorrhagic disease in Asian elephants. The data provided supports the authors’ conclusions, showing that viral replication is primarily associated with capillary endothelial cells, with no productive infection detected in other cell types. This finding reinforces the main conclusion regarding the virus's preference for endothelial cells.

2. The primary aim is to describe the tissue and cellular distribution of EEHV1A replication. The analysis is thorough within the context of its objectives and methods. Formal statistical analysis was not performed, which is acceptable and appropriate given the nature of the study. Additionally, considering the small and rare sample size (n = 12) and the nature of archival tissue-based pathology, the statistical power would be limited, and the inferential statistics would be of questionable value.

3. All relevant data are included in the manuscript. The authors provide detailed, case-by-case data in Tables 1-3, outlining which tissues were analysed and their corresponding ISH signal grades. The methods section clearly explains how the data were generated.

A few comments that could help strengthen the manuscript:

1. While formal inferential statistics are not essential, including summary statistics (e.g., mean ± SD, median, or range of ISH signal grades for each tissue) would enable readers to compare signal distributions across tissues more easily. Additionally, I recommend that the authors consider visualizing key results. Creating a bar chart or heat map showing the signal intensity grades across different tissue types could effectively summarize tropism patterns and highlight differences between the tissues more clearly.

2. The manuscript indicates that a single ACVP-certified pathologist reviewed the slides. While this contributes to consistency, it would be beneficial to note if a second reviewer confirmed the findings or if intra-observer reproducibility was evaluated.

3. While this study focuses on fatal EEHV1A-HD cases, a brief discussion on how the findings may relate to subclinical or early-stage infections could enhance the overall analysis and highlight the context-dependent nature of tropism.

4. The authors appropriately acknowledge that their study does not examine the early stages of infection, dissemination mechanisms, or latency, given that their samples are limited to fatal cases of EEHV1A-HD. Because the probes used in their research targeted genes associated with active viral replication (specifically, DNA polymerase and terminase), the authors should consider suggesting potential molecular targets for future studies. These could include alternative viral or host markers that may help in detecting latency or early infection, which are not addressed in this study.

**Do you want your identity to be public for this peer review?** For information about this choice, including consent withdrawal, please see our Privacy Policy

Reviewer #1: No

Reviewer #2: **Yes: ** Lawrence Annison

---

## [Author Response · Author response to Decision Letter 1]

6 Jul 2025

23 June 2025

PLOS One

Academic editors and peer reviewers

Dear Madam or Sir,

Thank you for the thoughtful consideration and suggested revisions of our manuscript titled, “Tissue and cellular tropism of elephant endotheliotropic herpesvirus (EEHV)1A in hemorrhagic disease” for publication as a Research Article in PLOS One. Please see below for our specific responses to all questions and comments.

Journal requirements

• Done

2. Please state what role the funders took in the study.

3. Please confirm at this time whether or not your submission contains all raw data required to replicate the results of your study.

• Confirmed. All relevant data is included. Raw data not included is the actual glass slides containing tissue sections assayed with in situ hybridization. Scanning all of the glass slides to create digital files and uploading these digital slide scans was not feasible and not deemed necessary to share the results of the study. Some additional images are included in the revised manuscript to address specific reviewer comments/queries.

4. Please remove any funding-related text from the manuscript and let us know how you would like to update your Funding Statement.

• Funding-related text has been removed from the manuscript Acknowledgments section.

• The current Funding Statement is correct as is: [The research was funded by a grant from the American Association of Zoo Veterinarians Wild Animal Health Fund (AAZV Proposal: 2022 #4).].

Reviewer #1

The title says "viral tissue" not sure what they mean with this term. My suggestion is to modify the title.

• The meaning is that the study examined the virus’s tropism for both host tissue types (e.g cardiac, hepatic, pulmonary) and specific host cell types (e.g. epithelial, endothelial, leukocyte). Title was revised to “Tissue and cellular tropism of elephant endotheliotropic herpesvirus (EEHV)1A in hemorrhagic disease” to enhance clarity per Reviewer’s request.

Although the pathology and tissue observation was made by a certified pathologist, the interpretation of data on the manuscript is difficult to follow. It might be better describe by tissue?

• The root of the Reviewer’s difficulties in understanding the results is uncertain. The Results currently begin by reporting that positive hybridization was only detected in endothelial cells in all cases (reporting cellular tropism), regardless of tissue. The Results then proceed to describe the relative amount of signal in each tissue type examined (reporting tissue tropism). The authors do not know what is meant by “describe by tissue”.

As the authors mention, signal grade was semi-quantitative, ¿were those based only on the pathologist criteria? because there´s no reference associated to this.

• The grading scheme employed to assess the relative amount of positive hybridization signal in each tissue is described in the Methods (line 154-157): “Signal was semi-quantitatively graded as follows: Grade 1 – 1-9 positive nuclei/200x magnification field; Grade 2 – 10-20 positive nuclei/200x field; Grade 3 – 21-30 positive nuclei/200x field; Grade 4 – greater than 30 positive nuclei/200x field.” This grading was performed to provide some assessment of signal amount (i.e viral load) in tissues, and the particular scheme was developed for use in the study; no applicable, previously-published grading scheme existed. A supplemental figure image (S1 Fig) has been added to the submission showing examples of grade 1-4 positive ISH signal.

Is there any data about the sensibility/specificity of the RNAscope 2.5 HD Detection Reagent-Red Kit.

• ISH sensitivity/specificity are features of the target hybridization probes, not the detection kit. The experimental probes were designed and generated for specific use in the study. Sensitivity and specificity were based on alignment with target gene sequence (GenBank Accession KC618527) submitted to Advanced Cell Diagnostics for probe design. Actual probe nucleic acid sequence is proprietary and not released by ACD though all designed probes become publicly available in the ACD product catalogue (probe part numbers [PN] are referenced in the manuscript). Sufficient numbers of known control EEHV1A infected samples do not exist to determine and calculate actual sensitivity and specificity; however, inherent features of the RNA Scope ISH protocol provide enhanced assay sensitivity and specificity (over conventional ISH methods). These include amplification of multiple, relatively short, adjacent target sequences and amplification of target signal via a cascade of hybridization events.

The authors need to include images of the negative controls used with the assay.

• A supplemental figure image (S2 Fig) of tissue hybridized with the negative probe has been added.

The authors need to include images of positive tissues to beta-actin.

• A supplemental figure image (S3 Fig) of tissue hybridized with the beta actin positive control probe has been added.

It is possible to count number of positive cells to beta actin and make some statistics with this number and those positive to the virus genes?

• For the purposes of this investigation, beta actin ISH was pursued to ensure tissue viability for downstream assay of EEHV1A genes of interest. All tissues from all cases had positive hybridization signal indicating tissues were viable for target gene ISH (i.e. formalin fixation of archival samples was not a significant impediment to target antigen retrieval). Beta actin detection was not done to try and normalize or equilibrate EEHV1A gene expression across samples. Beta actin expression and thus hybridization signal are not uniform across tissues or individuals. This observation was supported by the study results as indicated in lines 167-171: “Distribution and intensity of beta actin signal varied among cases and tissues. Cytoplasmic signal was common in vascular and alimentary tract smooth myocytes, neurons and glial cells, leukocytes and various epithelial cells. Nuclear signal was also sometimes prominent in cardiac, skeletal and smooth muscle and endothelial cells.”

The number of positive cells in each animal were similar? Let's say, in large intestine 8 samples out of 10 were considered positive grade 3, if they count 21-30 positive cells. But there's a difference of 9 cells between.

• The authors do not know what the Reviewer’s comment “the number of positive cells in each animal were similar?” is in reference to. The Results (lines 175-177) state: “Tissue and cellular distribution of hybridization signal was similar in all EEHV1A-HD cases, though variation in extent of tissue involvement and amount of signal within tissues was noted.” To document and examine variation among tissues and samples, grading of ISH signal was performed and reported. As stated, the grading system was semi-quantitative with ranges of 10 positive cells/200x field defining the sequential grades. To increase accuracy in grading, final grade was the average of individual counts of signal detected in ten 200x magnification fields (described in lines 153-154 of the Methods).

Is it possible to use these numbers to include some statistics?

• Given the small sample size (N=12) dictated by the limited number of total EEHV1A cases that have occurred in North America to date, statistical power was limited, and thus the aim of the current study was to provide descriptive information regarding the tissue and cellular tropism of the virus only; value of any inferential statistics was considered questionable.

Data on table 1 and 2, it is possible to present results in a more friendly/visual format? Table 3 repeats what is shown on table 2.

• Data from Table 3 has been used to produce a new figure to visually represent these results (Fig 1). The new figure replaces the original Table 3.

Authors mention that archival tissues were from PCR-confirmed Asian elephant cases of EEHV-HD due to EEHV1A. All tested tissues were positive to PCR?

• According to medical records, all cases were confirmed positive via PCR; confirmation was made on PCR of whole blood and/or tissues. Precise information of which tissues were tested and were positive for PCR in each case was not available for the majority of cases. Cases were archival, and many were from > 20 years ago.

What is the rationale to select the DNA polymerase and terminase genes to design the hybridization probes? Include also the sequence of each probe on the document.

• PCR assays utilized to detect EEHV viremia in elephants target the DNA polymerase and terminase probes, thus selection of these genes for use in the current study provided continuity with already established testing/methods. Additionally, because DNA polymerase and terminase are genes involved in viral replication, targeting of these genes was postulated to have increased sensitivity for detection of active disease (i.e hemorrhagic disease), the focus of the investigation. Actual probe nucleic acid sequence is proprietary and not released by ACD though all designed probes become publicly available in the ACD product catalogue.

The discussion takes the main aspects on the paper, including the novel kit used, the genes selected to target in the tissues. The main type of cells (endothelial) where positivity was detected.

An important aspect that should be discussed is the tissue treatment before in situ hybridization.

• Methods (lines133-137) described pretreatment of slides for ISH, ““Pretreatment solution 1” was applied to the slides for 10 minutes at room temperature. The slides were then boiled in “Pretreatment solution 2” at 100°C for 30 minutes, followed by protease digestion in “Pretreatment solution 3” for 30 minutes at 40°C to allow target accessibility.”

Reviewer #2

1. While formal inferential statistics are not essential, including summary statistics (e.g., mean ± SD, median, or range of ISH signal grades for each tissue) would enable readers to compare signal distributions across tissues more easily. Additionally, I recommend that the authors consider visualizing key results. Creating a bar chart or heat map showing the signal intensity grades across different tissue types could effectively summarize tropism patterns and highlight differences between the tissues more clearly.

• Data from Table 3 has been used to produce a new bar chart figure to visually represent these results (Fig 1). The new figure replaces the original Table 3 and hopefully highlights the differences between the tissues more readily.

2. The manuscript indicates that a single ACVP-certified pathologist reviewed the slides. While this contributes to consistency, it would be beneficial to note if a second reviewer confirmed the findings or if intra-observer reproducibility was evaluated.

• No additional pathologist was available for review of ISH light microscopy. Unfortunately, the other manuscript authors do not have expertise in histologic evaluation. To optimize accuracy in assignment of ISH grading, the protocol involved examination of ten 200x fields/tissue to count the number of endothelial nuclei with positive ISH signal, and then these were averaged for each tissue/case to assign the grade.

3. While this study focuses on fatal EEHV1A-HD cases, a brief discussion on how the findings may relate to subclinical or early-stage infections could enhance the overall analysis and highlight the context-dependent nature of tropism.

• Because the study focused on archival, fatal cases of hemorrhagic disease and, for viral detection, targeted viral genes involved in replication (and thus reflective of active/productive infection), results failed to shed light on other, early stages of infection or subclinical disease. A couple points in the Discussion attempt to address the limited application of the study. See lines 271-278: “findings of the current study show that clinical hemorrhagic disease is characterized exclusively by viral replication within and consequent damage to capillary endothelial cells. Evidence of epithelial invasion/involvement and/or dissemination via leukocyte trafficking was not detected in the cases of active EEHV-HD. In the future, studies providing specific detection of EEHV DNA genomes within cells in the absence of viral lytic gene RNA or protein expression might clarify the proportion of nonproductively infected cells to cells with ongoing active replication and further elucidate viral tropism.” And lines 313-319, “Given that positive ISH signal was only detected in capillary endothelial cells in all evaluated samples, the current study was unable to elucidate mechanisms of pathogenesis such as initial infection, dissemination, shedding and latency. Capillary endothelial cell infection and replication appeared to be vital features of the hemorrhagic disease stage of EEHV1A infection. At this stage of the disease and in these fatal infections, no ISH signal distribution compatible with active viral shedding or sites of latency were observed.” Despite the lack of direct applications, the study results are valuable in establishing a foundation for additional work. Results provide validation for use of RNA Scope ISH for detection of EEHV in elephant tissues and demonstrate viability of the application even in archival samples.

4. The authors appropriately acknowledge that their study does not examine the early stages of infection, dissemination mechanisms, or latency, given that their samples are limited to fatal cases of EEHV1A-HD. Because the probes used in their research targeted genes associated with active viral replication (specifically, DNA polymerase and terminase), the authors should consider suggesting potential molecular targets for future studies. These could include alternative viral or host markers that may help in detecting latency or early infection, which are not addressed in this study.

• Thank you. A specific herpesviral gene target encoding for a structural protein that has been utilized for EEHV detection in previous studies has been added to the pertinent section of the Discussion, lines 325-328: “Future studies surveying elephant tissues from survivors of previous EEHV-HD and/or healthy, infected adults and detecting other non-replicative gene targets (e.g. viral glycoprotein B) may provide further information.”

---

## [Decision Letter · Decision Letter 1]

5 Aug 2025

Tissue and cellular tropism of elephant endotheliotropic herpesvirus (EEHV)1A in hemorrhagic disease

PONE-D-25-19109R1

Dear Dr. Landolfi,

We’re pleased to inform you that your manuscript has been judged scientifically suitable for publication and will be formally accepted for publication once it meets all outstanding technical requirements.

Kind regards,

Graciela Andrei

Academic Editor

PLOS ONE

Additional Editor Comments (optional):

Reviewers' comments:

Reviewer's Responses to Questions

**Comments to the Author**

Reviewer #1: All comments have been addressed

Reviewer #2: All comments have been addressed

2. Is the manuscript technically sound, and do the data support the conclusions?

Reviewer #1: Yes

Reviewer #2: (No Response)

3. Has the statistical analysis been performed appropriately and rigorously?

Reviewer #1: N/A

Reviewer #2: (No Response)

4. Have the authors made all data underlying the findings in their manuscript fully available?

Reviewer #1: Yes

Reviewer #2: (No Response)

5. Is the manuscript presented in an intelligible fashion and written in standard English?

Reviewer #1: Yes

Reviewer #2: (No Response)

Reviewer #1: All comments have been addressed. This is a very interesting work that reports important data on viral diseases poorly known in animal populations.

Reviewer #2: (No Response)

**Do you want your identity to be public for this peer review?** For information about this choice, including consent withdrawal, please see our Privacy Policy

Reviewer #1: No

Reviewer #2: **Yes: ** Lawrence Annison

---

## [Editor Report · Acceptance letter]

PONE-D-25-19109R1

PLOS ONE

Dear Dr. Landolfi,

I'm pleased to inform you that your manuscript has been deemed suitable for publication in PLOS ONE. Congratulations! Your manuscript is now being handed over to our production team.

Kind regards,

on behalf of

Dr. Graciela Andrei

Academic Editor

PLOS ONE